# Food-insecure pregnant women in South Africa: a cross-sectional exploration of maternal depression as a mediator of violence and trauma risk factors

Whitney Barnett,[1,2] Jennifer Pellowski,[3] Caroline Kuo,[4] Nastassja Koen,[5,6] Kirsten A Donald,[7] Heather J Zar,[1,2] Dan J Stein[5,6]

For numbered affiliations see end of article.

**Correspondence to**
Mrs Whitney Barnett;
barnett.whitney@gmail.com

## ABSTRACT

**Objectives** Better understanding of psychosocial risk factors for food insecurity (FI) during pregnancy and how they interact is crucial, given long-term health implications for maternal and child health. We investigated the association between maternal childhood trauma as well as intimate partner violence (IPV) and FI among pregnant women in South Africa, in the Drakenstein Child Health Study, and whether maternal depression mediates these relationships.

**Setting** Two primary care clinics in Paarl, South Africa.

**Participants** 992 pregnant women; inclusion criteria were clinic attendance and remaining in area for at least 1 year; women were excluded if a minor.

**Methods** We examined psychosocial predictors of FI using multivariate regression. Mediation analyses investigated whether depression mediated the relationship between IPV and FI as well as between childhood trauma and FI, including disaggregation by two study communities. FI was assessed using an adapted US Department of Agriculture food security scale; households were coded as food insecure where 2 of 5 affirmative responses were recorded.

**Results** Among 992 pregnant women, there were high rates of IPV (7%–27%), depression (24%) and childhood trauma (34%). In multivariate cross-sectional analysis, emotional IPV (adjusted OR [aOR] 1.60; 95% CI 1.04 to 2.46), depression (aOR 1.05; 95% CI 1.01 to 1.08) and childhood trauma (aOR 1.52; 95% CI 1.08 to 2.15) predicted FI. In mediation models, depression partially mediated the relationship between emotional IPV and FI as well as physical IPV and FI; depression partially mediated the relationship between childhood trauma and FI. Differing degrees of mediation were found when applied to communities.

**Conclusions** Antenatal maternal depression, IPV and childhood trauma were highly prevalent and associated with FI. Depression, IPV and trauma screening services should be considered within routine antenatal care and may offer an opportunity to identify and intervene. Community-level differences in risk and in mediation analyses indicate that contextual tailoring of interventions may be important.

### Strengths and limitations of this study

► There are few studies investigating depression as a mediator in relationships between subtypes of maternal intimate partner violence or childhood trauma and food insecurity during pregnancy.
► This study extends existing related research to a low-resource African population with a large sample size.
► The current study was a cross-sectional analysis; therefore, further research is needed to assess the direction of causality, and if differences exist by trimester and postpartum.

## BACKGROUND

Food insecurity is the lack of nutritionally adequate and safe food or a limited ability to acquire necessary food in socially acceptable ways.[1] The Food and Agriculture Organization estimates that 689 million people worldwide (1 in 10) suffer from severe food insecurity (2014–2016); Africa has the highest prevalence of severe food insecurity (27.4%), almost four times the prevalence of other regions.[2] Studies have shown a link between food insecurity and poor pregnancy outcomes, including low birth weight,[3] gestational diabetes and pregnancy complications.[4] In addition, young children in food-insecure households have poorer general health,[5–7] increased probability of being hospitalised,[6 8] lower levels of parent–child attachment[9] and increased developmental delays.[9–11] Chronic hunger in childhood has also been linked to a higher likelihood of chronic medical conditions, such asthma, heart conditions, kidney disease or allergies.[12] Pregnant women may be particularly vulnerable to food insecurity

due to increased nutrient demands and the inability to continue working, leading to financial strain.

Maternal mental health disorders are prevalent in low-income and middle-income countries (LMICs). Maternal mental health problems such as depression[13] and psychosocial risk factors such as stressful life events, intimate partner violence (IPV) and trauma[14–17] are associated with food insecurity as well as poorer pregnancy outcomes such as low infant birth weight,[18] impaired fetal[19] and infant growth and nutritional status[3 20] as well as poorer infant cognitive development.[8 21] Although the relationship between maternal trauma or violence exposures as well as mental health and food insecurity has been explored, few studies have investigated depression as a mediator in the relationship between other psychosocial risk factors (eg, violence or trauma) and food insecurity. Sun and colleagues,[22] in a large US based study, found maternal childhood trauma to be linked to food insecurity during pregnancy and that depression mediated this relationship. Others in the USA have found similar links between childhood trauma and food insecurity but have not investigated mental health pathways.[23] In another US-based study, IPV was found to be a significant predictor of food insecurity, mediated by depression.[24] However, this study did not find differential associations between subtypes of IPV (emotional, physical and sexual), though others have.[25] The majority of studies have focused on high-income countries[14 16 22 24 26] or have used small sample sizes to explore associations.[14 23] The current study aims to extend previous research to an LMIC context using a large study sample and to analyse multiple exposures, maternal trauma, IPV and stressful events, which are often co-occuring and have a higher prevalence in LMIC settings.

Examining maternal psychosocial risk factors and mental health characteristics in relation to food insecurity in LMICs is important. Particularly in the context of high proportions of maternal headed and single parent households and given the high prevalence of maternal psychosocial risk factors, especially during pregnancy, when exposures can adversely affect both maternal and child long-term health. Food security is a managed process such that family members have some control over how they cope with food insecurity and who within the family experiences it.[26 27] This ability to manage the effects of food insecurity may be adversely affected by maternal psychosocial risk factors and maternal mental health.[16] Furthermore, community-level factors such as differences in stigma, access to care, gender norms affecting agency or education levels for women, may have significant differential effects within communities. This community context may be important to understand how to best address key risk factors for food insecurity and to inform design of effective interventions.

We therefore aim to explore associations between maternal psychosocial risk factors or mental health and food insecurity, disaggregated by two communities with different risk profiles and community-level factors. We

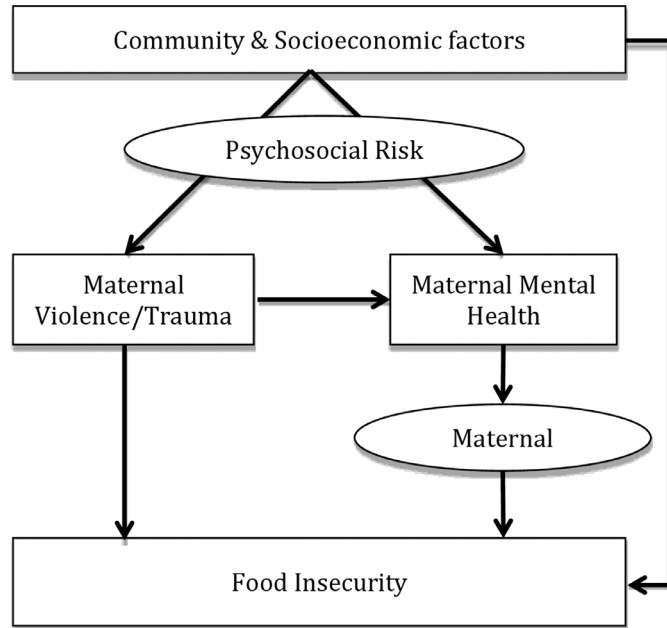

**Figure 1** Conceptual framework for study.

investigate whether depression acts as a mediator in the relationship between IPV or childhood trauma and food insecurity in a LMIC context (see figure 1). This extends the evidence base to geographic regions where these issues are highly prevalent, but the relationships between these variables are poorly understood and rarely investigated.

Given the long-term health implications of food insecurity for child development as well as maternal and child mental and physical health,[12–15] understanding how risk factors for poor child health outcomes interact is critical to inform public policy to address the most urgent modifiable risk factors. Previous published findings from this cohort have shown links between psychosocial risk factors and food insecurity during pregnancy;[28] this paper builds on that research by investigating the mediational effects of maternal depression on the relationship between emotional, physical and sexual IPV and food insecurity as well as maternal childhood trauma and food insecurity. Addressing food insecurity during pregnancy offers an opportunity to link antenatal care with nutritional programmes and manage associated mental health risk factors at a time when those risk factors impact the mother's safety and well being as well as infant outcomes after birth.

## METHODS
The Drakenstein Child Health Study (DCHS) is a multidisciplinary population-based birth cohort study located in a periurban area, 60 km outside of Cape Town, South Africa. It is a low socioeconomic community comprising approximately 200 000 people, predominantly of mixed-ancestry (62.5%; 13.5% Caucasian; 22.7% black African).[29] The district is characterised by a high prevalence of a range of health risk factors such as single-parent households, depression, childhood trauma, IPV, poverty,

low levels of education (27.4% completing secondary school) and high unemployment (17.6%). The DCHS is a longitudinal cohort study following mother–child dyads through early childhood.[30 31] The current analysis uses data from two antenatal visits: maternal psychosocial health and food security were measured at an antenatal visit between 28 and 32 weeks' gestation; sociodemographics were measured at the enrolment visit, at 20–28 weeks' gestation.

## Participants

Pregnant women were enrolled from March 2012 to March 2015. Women were enrolled in their second trimester, between 20 and 28 weeks' gestation at two public sector primary healthcare clinics, one serving a predominantly mixed-ancestry population (TC Newman) and the other serving a predominantly black African population (Mbekweni). Inclusion criteria were: (1) attendance at one of the two study clinics and (2) intending to remain in the study area for at least 1 year. Mothers were excluded if they were under 18 years of age at enrolment or were not pregnant.

## Measures

Maternal sociodemographics and mental health was measured using validated questionnaires administered by trained study staff across two antenatal visits. Mental health assessments included measures of IPV, depression, childhood trauma, stressful life events and psychological distress. The IPV Questionnaire used in this study was adapted from the WHO multicountry study[32] and the Women's Health Study in Zimbabwe.[33] Participants were dichotomised into exposed or unexposed for having experienced emotional, physical or sexual IPV in the past 12 months; exposure was defined as a score >1 indicating more than an isolated incident within each subtype. The Edinburgh Postnatal Depression Scale (EPDS)[34] was used to measure depression; this scale has been validated for use with pregnant women and in a South African population.[35 36] The EPDS consists of 10 items referring to the past 7 days with each item assessed on a scale from 0 to 3. A total score was obtained by summing responses for all items and was included as a continuous score, with higher scores indicating more severe depressive symptoms; total scores were included in models. To present baseline cohort characteristics depression was dichotomised using a cut-off score of ≥13 to classify women as depressed.[34] The Childhood Trauma Questionnaire (CTQ)[37] Short-Form was used to assess abuse and neglect experienced as a child. Each item was responded to on a five-point scale ranging from *1=never true* to *5=very often true*. Continuous scores were used with a total possible range from 28 to 140. Where dichotomised, a cut-off score of >36 was used to indicate exposure to childhood trauma, as described in the CTQ manual.[38] The Modified World Mental Health Life Events Questionnaire, adapted based on items used in the South African Stress and Health Study (SASH) in South Africa,[39] was used to measure stressful or negative life events in the past year (eg, serious illness, major financial crisis and serious discord with family or friends). Items were scored according to whether the event was experienced, *0=no, 1=yes*. Individual items were then summed to create a total score, ranging from 0 to 17, with higher scores indicating greater exposure to stressful life events. Dichotomous exposure to stressful life events was defined as experiencing at least one such event. The SRQ-20 is a WHO-endorsed measure of psychological distress.[40] The SRQ-20 consists of 20 items, which assess non-psychotic symptoms, including symptoms of depressive and anxiety disorders, scored according to whether the symptom was present, *0=no, 1=yes*. Individual items are summed to generate a total score ranging from 0 to 20, with higher scores indicating higher levels of psychological distress.[41] A cut-off score of ≥8 was used to classify participants into high versus low risk, as has been used elsewhere.[41 42]

Sociodemographic variables including mother-reported household factors and maternal demographics were collected using an interviewer-administered questionnaire adapted from items used in the SASH Study.[39] Socioeconomic status (SES) was measured based on a composite score of asset ownership, household income, employment and education.[39] Social grants (receiving government support for child care or disability) were self-reported by mothers at enrolment.

Perceived food insecurity was assessed using an adapted version of the US Department of Agriculture Short Form Household Food Security Scale,[1] which captures food hardship due to financial constraints. Specific questions asked about whether meals were made smaller for children in home, whether children skipped meals or went hungry and whether children in home went a full day without eating—due to limited financial means within the home, as described previously.[28] Questions included referred to children in the home as a conservative estimate of perceived food insecurity; studies have shown that parental buffering often means that children are the last household members to experience food insecurity.[43 44] Five items were used, and an affirmative response to two or more items was coded as being food insecure.

## Ethics

Mothers gave written informed consent at enrolment.

## Patient and public involvement

Prior to study initiation, local stakeholders (Department of Health staff and managers) were involved in the planning of the parent study, the Drakenstein Child Health birth cohort study. Patients and public were not involved in conceptualisation or analysis of the specific aims reported in the current study; however, study findings are routinely fed back to the study community.

## STATISTICAL ANALYSIS

All data were analysed using IBM SPSS V.22. Univariate logistic regression analyses were conducted to determine

the bivariate relationship between food insecurity and demographic and psychosocial predictors. ORs with p values were calculated to determine the strength of these associations. A hierarchical multivariate logistic regression analysis was conducted to independently evaluate IPV exposure and trauma/stress on food insecurity prior to the addition of mental health risk factors while controlling for demographic variables. Block 1 included community, maternal income and maternal education. Block 2 included recent experiences of emotional, physical and sexual IPV as well as maternal childhood trauma and stressful life events. Finally, block 3 added depression and psychological distress. To determine whether depression played a mediating role on the relationship between IPV and food insecurity, mediational analyses were conducted using PROCESS macro.[45] Model number 4 was used and indirect effects were bootstrapped using 1000 samples. Beta coefficients and SEs are reported for all paths and 95% CIs are reported for the indirect effects. Models were conducted for the full sample and then for each community individually; models were split by community because of the socioeconomic, cultural, clinical and psychosocial differences between the two communities that could have significant bearings on the results of the mediation models. This process was replicated for depression as a mediator of the relationship between childhood trauma and food insecurity. Mediation models controlled for community, maternal income, maternal education, social grants, number of children in the household and HIV status; childhood trauma was controlled for in all IPV mediation models, and emotional, physical and sexual IPV were controlled for in the childhood trauma mediation models.

## RESULTS

A total of 1225 pregnant women were enrolled between March 2012 and February 2015; of these, 992 women had complete data and were included in the analysis. Missing data resulted from non-attendance at the second antenatal visit where psychosocial data were collected. A sensitivity analysis was therefore only done on sociodemographic variables (clinic, education, income, employment, social grants and whether married); those mothers included in the present analysis versus the whole cohort differed significantly only regarding whether mothers received social grants (online supplementary table 1). Detailed baseline demographic characteristics, stratified by recruitment site, are presented in table 1. The median age of participants was 26.6 years (SD 5.8). The sample was characterised by low SES—77% of mothers had a monthly income of less than R1000 (approximately US$100), 49% of mothers were receiving social assistance, 26% reported being employed and 38% completed secondary education (high school). A minority of mothers (40%) were married or with a partner. Food security, HIV prevalence and SES quartiles were significantly different between clinics as were the majority of psychosocial variables.

Households in Mbekweni were much more likely to be food insecure than households at TC Newman (45.7% vs 12.6%). Mothers at TC Newman were significantly more likely to have experienced emotional and sexual past year IPV as well as childhood trauma and stressful life events. Co-occurrence of mental health issues was prevalent, though more so at TC Newman. Overall, 12% of mothers had both depression and IPV, 13% depression and childhood trauma and 16% childhood trauma and any form of IPV.

In bivariate analysis (table 2), antenatal food insecurity was significantly more likely among participants from Mbekweni (adjusted OR [aOR] 5.82; 95% CI 4.20 to 8.07), those who had not completed secondary school (aOR 0.43; 95% CI 0.32 to 0.57), mothers with lower income levels (aOR 0.39; 95% CI 0.27 to 0.56), mothers who had experienced emotional IPV (aOR 1.44; 95% CI 1.07 to 1.94) or physical IPV (aOR 1.84; 95% CI 1.35 to 2.52) in the past 12 months and mothers with higher levels of antenatal depression (aOR 1.09; 95% CI 1.06 to 1.12), childhood trauma (aOR 1.49; 95% CI 1.13 to 1.97) and psychological distress (aOR 1.04; 95% CI 1.01 to 1.08).

### Hierarchical regression

A hierarchical logistic regression was done to investigate the additive impact of risk factor groups on food security (table 2). Throughout all blocks, community, maternal education and maternal income remained significantly associated with food insecurity. In block 2 among IPV, trauma and stress risk factors, adjusting for maternal sociodemographic factors and community, emotional IPV and childhood trauma were significant predictors of food insecurity. In the final model (block 3), which incorporated all psychological variables and demographic variables, mothers from Mbekweni were almost eight times (aOR 7.85; 95% CI 5.29 to 11.66) as likely as TC Newman mothers to experience antenatal food insecurity. Mothers who completed secondary school were 54% less likely to experience food insecurity (aOR 0.46; 95% CI 0.33 to 0.64) compared with mothers who did not complete secondary school. Similarly, mothers with higher incomes were 56% (aOR 0.44; 95% CI 0.29 to 0.67) less likely to experience food insecurity. Mothers who experienced emotional IPV in the past 12 months were 60% more likely (aOR 1.60; 95% CI 1.04 to 2.46), mothers with higher depression scores on EPDS were 5% more likely (aOR 1.05; 95% CI 1.01 to 1.08) and mothers with a history of childhood trauma were 52% more likely (aOR 1.52; 95% CI 1.08 to 2.15) than mothers without these psychological risk factors to experience food insecurity.

### Depression as a mediator of the relationship between IPV and food insecurity

In mediation models including both communities (figure 2A), depression partially mediated the relationship between emotional IPV and food insecurity (direct effect p value=0.0001; indirect effect=0.16, 95% CI

**Table 1** Maternal demographic and psychological variables

| | Overall n (%) | TC Newman n (%) | Mbekweni n (%) | $X^2$ | P value |
|---|---|---|---|---|---|
| Number of mothers | 992 | 443 | 549 | | |
| Mean age of mother (SD) | 26.6 (5.8) | 25.7 (5.4) | 27.3 (5.9) | –4.543* | *** |
| Food insecurity | | | | | |
| Secure | 685 (69.1) | 387 (87.4) | 298 (54.3) | 125.53 | *** |
| Insecure | 307 (30.9) | 56 (12.6) | 251 (45.7) | | |
| Race | | | | | |
| Black | 548 (55) | 6 (1) | 542 (99) | 943.05 | *** |
| Coloured | 443 (45) | 437 (99) | 6 (1) | | |
| SES quartiles | | | | | |
| Lowest SES | 258 (26) | 81 (18) | 177 (32) | 37.27 | *** |
| Low to moderate SES | 261 (26) | 117 (26) | 144 (26) | | |
| Moderate to high SES | 242 (24) | 109 (25) | 133 (24) | | |
| Highest SES | 231 (23) | 136 (31) | 95 (17) | | |
| Maternal income | | | | | |
| <R1000/month | 767 (77) | 330 (74) | 437 (80) | 7.86 | * |
| R1000–R5000/month | 212 (21) | 103 (23) | 109 (20) | | |
| R5000–R10 000/month | 12 (1) | 9 (2) | 3 (1) | | |
| Receive social assistance | 491 (49) | 221 (50) | 270 (49) | 0.085 | 0.798 |
| Maternal education | | | | | |
| Some secondary | 613 (62) | 266 (60) | 347 (63) | 1.037 | 0.308 |
| Completed secondary | 379 (38) | 177 (40) | 202 (37) | | |
| Median number of children in household | 1 | 1 | 1 | 22.191 | ** |
| Married/cohabiting | 399 (40) | 200 (45) | 199 (36) | 10.064 | * |
| Employed | 254 (26) | 132 (30) | 122 (22) | 7.439 | ** |
| Maternal HIV | 216 (22) | 17 (4) | 199 (36) | 151.195 | *** |
| Psychosocial risk factors | | | | | |
| Past year IPV | | | | | |
| Emotional IPV | 266 (27) | 155 (35) | 111 (20) | 27.26 | *** |
| Physical IPV | 216 (22) | 106 (24) | 110 (20) | 2.18 | 0.14 |
| Sexual IPV | 68 (7) | 49 (11) | 19 (3) | 22.179 | *** |
| Probable depression (EPDS ≥13) | 242 (24) | 112 (25) | 130 (24) | 0.341 | 0.559 |
| Childhood trauma | 335 (34) | 179 (40) | 156 (28) | 15.761 | *** |
| Psychological distress | 208 (21) | 109 (25) | 99 (18) | 6.39 | 0.011 |
| Stressful life events | 449 (45) | 265 (60) | 184 (34) | 68.467 | *** |
| Co-occurrence of psychosocial risk factors | | | | | |
| Depression and any IPV | 122 (12) | 68 (15) | 54 (10) | 6.911 | ** |
| Depression and childhood trauma | 124 (13) | 76 (17) | 48 (9) | 15.864 | *** |
| Childhood trauma and any IPV | 154 (16) | 101 (23) | 53 (10) | 32.304 | *** |

Note: psychological risk factors listed where above threshold; IPV above threshold=score of >1 within each subtype (mothers experiencing more than an isolated incidence in past year); depression above threshold=score ≥13; childhood trauma above threshold where score >36; psychological distress dichotomised into low and high risk categories where high risk=score ≥8; stressful life events presented where greater than 1.
*p<0.05, **p<0.01, ***p<0.001.
EPDS, Edinburgh Postnatal Depression Scale; IPV, intimate partner violence.

0.07 to 0.29) and partially mediated the relationship between physical IPV and food insecurity (direct effect p value=0.001; indirect effect=0.17, 95% CI 0.07 to 0.28). Sexual IPV was not tested in a mediation model because the bivariate relationship between sexual IPV and food insecurity was not significant (OR 1.62; p=0.06). Mediation models were split by community due to the high significance of recruitment community

**Table 2** Hierarchical logistic regression of variables associated with food insecurity

| Variables | Unadjusted OR (95% CI) | P value | Block 1 Adjusted OR (95% CI) | P value | Block 2 Adjusted OR (95% CI) | P value | Block 3 Adjusted OR (95% CI) | P value |
|---|---|---|---|---|---|---|---|---|
| *Demographic variables* | | | | | | | | |
| Community | 5.82 (4.20 to 8.07) | *** | 6.02 (4.30 to 8.41) | *** | 8.22 (5.60 to 12.06) | *** | 7.85 (5.29 to 11.66) | *** |
| Maternal income | 0.39 (0.27 to 0.56) | *** | 0.42 (0.28 to 0.62) | *** | 0.44 (0.29 to 0.66) | *** | 0.44 (0.29 to 0.66) | *** |
| Maternal education | 0.43 (0.32 to 0.57) | *** | 0.42 (0.31 to 0.58) | *** | 0.45 (0.32 to 0.63) | *** | 0.46 (0.33 to 0.64) | *** |
| *Intimate partner violence (IPV)* | | | | | | | | |
| Emotional IPV | 1.44 (1.07 to 1.94) | * | | | 1.67 (1.09 to 2.56) | * | 1.60 (1.04 to 2.46) | * |
| Physical IPV | 1.84 (1.35 to 2.52) | *** | | | 1.41 (0.91 to 2.18) | 0.121 | 1.32 (0.85 to 2.05) | 0.216 |
| Sexual IPV | 1.62 (0.98 to 2.68) | 0.061 | | | 1.77 (0.92 to 3.39) | 0.085 | 1.50 (0.78 to 2.89) | 0.253 |
| *Trauma/stress†* | | | | | | | | |
| Childhood trauma | 1.49 (1.13 to 1.97) | ** | | | 1.66 (1.18 to 2.33) | ** | 1.52 (1.08 to 2.15) | * |
| Stressful life events | 0.96 (0.90 to 1.02) | 0.157 | | | 0.98 (0.91 to 1.06) | 0.585 | 0.93 (0.86 to 1.01) | 0.089 |
| *Mental health†* | | | | | | | | |
| Depression (EPDS) | 1.09 (1.06 to 1.12) | *** | | | | | 1.05 (1.01 to 1.08) | ** |
| Psychological distress | 1.04 (1.01 to 1.08) | * | | | | | 1.05 (1.00 to 1.10) | 0.080 |
| | | | Block X² (df) | p-value | Block X² (df) | P value | Block X² (df) | P value |
| | | | 188.93 (3) | *** | 8.44 (2) | * | 15.75 (2) | *** |

*p<0.05, **p<0.01, ***p<0.001.
†Trauma/stress and mental health variables were included as continuous scores in regression analyses.
EPDS, Edinburgh Postnatal Depression Scale.

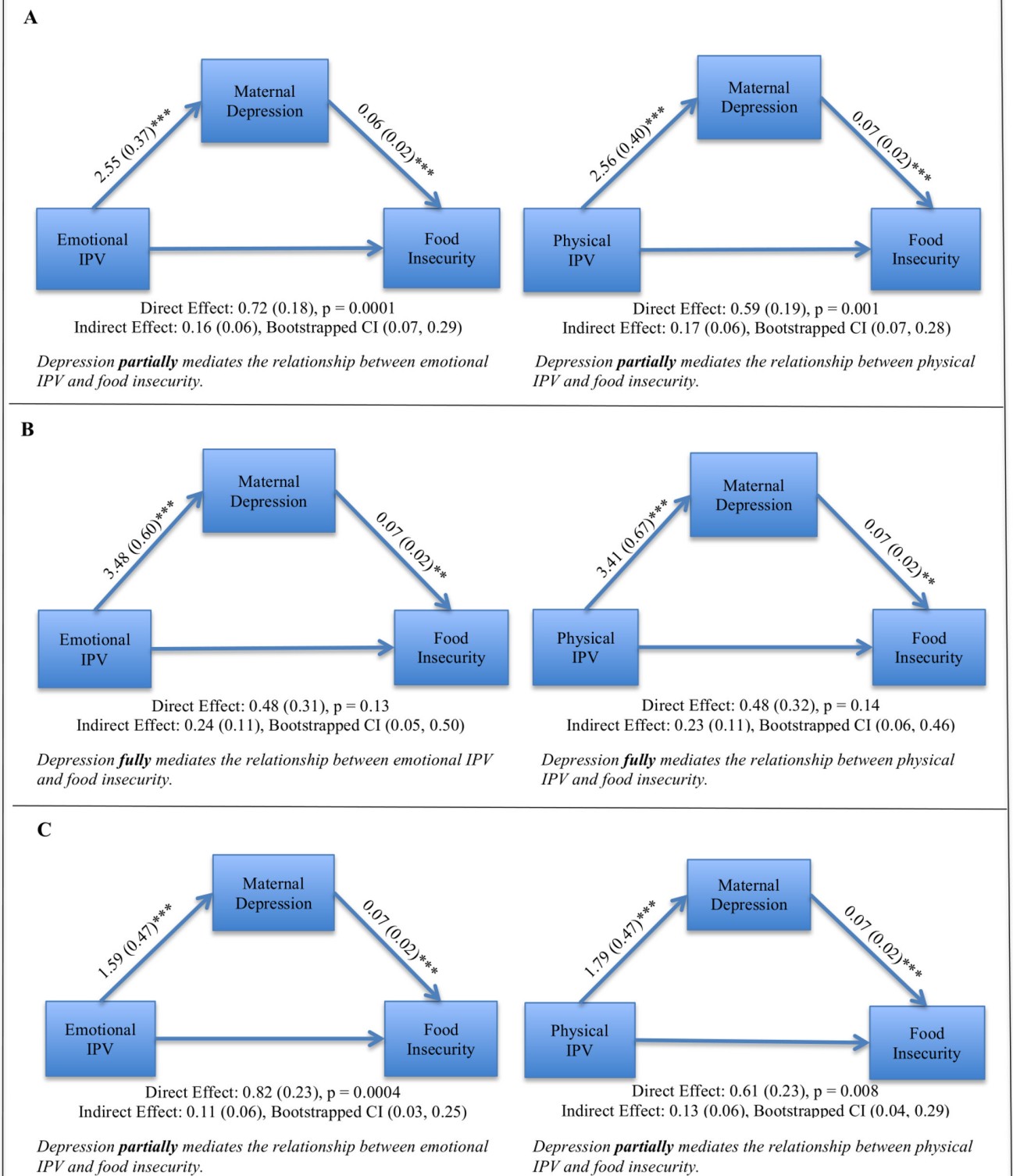

**Figure 2** Mediation models investigating depression as a mediator for physical IPV as well as emotional IPV and food insecurity. Covariates included in all models are: community, maternal income, maternal education, social grants, number of children in the household, HIV status and childhood trauma. (A) Combined community meditation models, n=992. (B) TC Newman mediational models, n=443. (C) Mbekweni mediational models, n=549. **p < 0.01 , ***p < 0.001.

as an independent predictor of food insecurity; when split by community, differing degrees of mediation were found. At TC Newman (figure 2B), depression fully mediated the relationship between emotional IPV

and food insecurity (direct effect p value=0.13; indirect effect=0.24, 95% CI 0.05 to 0.50) as well as between physical IPV and food insecurity (indirect effect=0.23, 95% CI 0.06 to 0.46). At Mbekweni (figure 2C), depression

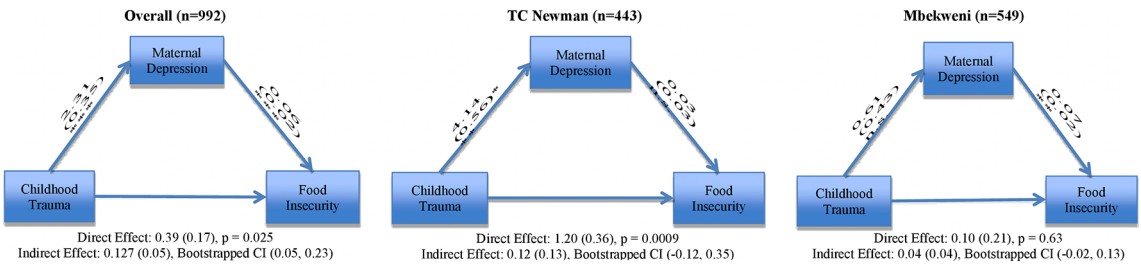

**Figure 3** Mediation models for both communities, for TC Newman and for Mbekweni investigating depression as a mediator for childhood trauma and food insecurity. Depression partially mediated the overall relationship between childhood trauma and food insecurity but did not mediate the relationship at TC Newman and did not mediate the relationship at Mbekweni. Covariates included in the model are: community, maternal income, maternal education, social grants, number of children in the household, HIV status and emotional, physical and sexual forms of IPV. IPV, intimate partner violence.

partially mediated the relationship between emotional IPV and food insecurity (direct effect p value=0.0004; indirect effect=0.11, 95% CI 0.03 to 0.25) and partially mediated the relationship between physical IPV and food insecurity (direct effect p value=0.008; indirect effect=0.13, 95% CI 0.04 to 0.29).

### Depression as a mediator of the relationship between childhood trauma and food insecurity

In mediation models including both communities (figure 3), depression partially mediated the relationship between childhood trauma and food insecurity (direct effect p value=0.025; indirect effect=0.13, 95% CI 0.05 to 0.23). These mediation models were also split by community, due to the high significance of community as a risk factor for food insecurity. When applying mediation models to childhood trauma at TC Newman, depression did not mediate the relationship between childhood trauma and food insecurity (direct effect p value=0.0009; indirect effect 0.12, 95% CI –0.12 to 0.35). Additionally, at Mbekweni, depression did not mediate the relationship between childhood trauma and food insecurity (direct effect p value=0.63; indirect effect 0.04, 95% CI –0.02 to 0.13).

### DISCUSSION

Our goal was to investigate the association between IPV or maternal childhood trauma and food insecurity during pregnancy, as well as to investigate maternal depression as a mediator for these relationships in an LMIC country, South Africa. We found significant effects of emotional IPV and maternal childhood trauma on antenatal food insecurity, after adjusting for community, maternal income and education. Mothers experiencing emotional IPV or with a history of childhood trauma were 60% and 52% more likely, respectively, to live in food insecure households while pregnant. Though previous studies have investigated links between IPV or childhood trauma and food insecurity, the current study extends this research to a low-resource setting with a large sample size.

Hernandez and colleagues[24] found that IPV was a significant predictor of food insecurity and that this was mediated by depression. However, this US-based study did

not find significant associations between subtypes of IPV and food insecurity; only a composite measure of IPV was found to be significant. Our research found that subtypes of IPV were differentially associated with food insecurity, with emotional IPV the only significant predictor in the final model. While mediation models split by site found a mediational effect of depression on this relationship at both clinics; emotional IPV did maintain a direct effect on food insecurity at Mbekweni. This may be an important distinction when planning effective interventions that consider community contexts; qualitative research has found that women feel emotionally abusive acts are more devastating than physical violence.[46] Emotional IPV, therefore, may be a critical and often overlooked risk factor for food insecurity. Furthermore, emotional IPV may manifest differently in an LMIC setting, compared with a high-income setting, where traditional gender norms may affect women's sense of power and identity and therefore, compounded by potential mental health sequelae, may further decrease her ability to manage household resources.[47]

Maternal childhood trauma also emerged as a critical risk factor for food insecurity during pregnancy. Sun and colleagues investigated this link in a large US-based study.[22] These authors reported that childhood trauma was linked to food insecurity during pregnancy and found that maternal depression modified this relationship. They found a dose–response relationship between number of childhood adverse events and severity of food insecurity; when considered together with depression, there was a greater impact on food insecurity. While other high-income country studies have also investigated this association, these have been limited by small sample sizes (n=44, n=31).[48 49] To our knowledge, our study is the first to investigate the link between childhood adversity and food security in an LMIC or mental health as a mediator in this relationship. Childhood trauma measured by family instability, violence exposure at a young age, and food insecurity or neglect in childhood is associated with many of the known risk factors of current food insecurity such as lower levels of education, employment and poor mental health outcomes in adulthood.[50] This highlights a critical link between childhood experiences and adult outcomes and

the intergenerational effects of trauma. In the context of maternal mental health, this may be particularly relevant as maternal hardship and stress may increase the likelihood of a traumatic childhood for their offspring. As noted by Sun and colleagues,[22] there is an intergenerational transmission of disadvantage, which highlights the need for a multifaceted approach to address food insecurity. Our findings reveal important intergenerational associations between food insecurity and maternal childhood exposure to violence and suggest that future research is needed to understand how intergenerational transmission of trauma occurs between mothers and children and what can be done to break this cycle. In high-prevalence settings in particular, intervention programmes should offer more than nutrition support and should include trauma-informed mental health services to reduce the transmission of trauma from one generation to the next, though further study is needed to determine if trauma counselling or interventions may help to alleviate the prevalence of food insecurity.

The co-occurrence of psychosocial risk factors was high (12%–16%) in our study sample. In order to better understand how these risk factors influence one another, we investigated depression as a mediator in these relationships. In overall models, depression partially mediated the relationship between emotional and physical IPV and food insecurity. Notably, the degree to which depression mediated this relationship differed between clinic communities. Depression fully mediated the relationship between emotional and physical IPV and food insecurity at TC Newman, but only partial mediation was found at Mbekweni. This highlights depression as important in the pathway through which IPV affects food insecurity at TC Newman. However, at Mbekweni, though depression also exacerbates this relationship, there may be other factors that explain the significant relationship between IPV and food insecurity. While maternal income was controlled for in mediation models, SES quartiles indicate that Mbekweni mothers are economically worse off than TC Newman mothers. This may be impeding the process of food management at Mbekweni, especially in the context of IPV. In overall models investigating depression as a mediator for the relationship between childhood trauma and food security, partial mediation was again seen; however, no mediation was found when models were split by community. At TC Newman, childhood trauma maintained a significant direct effect on food insecurity. It may be that social support networks are more robust at Mbekweni, thus mitigating the downstream effects of childhood trauma in that community. A study in a similar community in South Africa found that social support buffers the effect of trauma on depression symptoms[51]; further,more social support has been found to be particularly important for women, compared with men in mitigating mental health outcomes such as depression.[52] Additional research is needed to understand how or why depression mediated the effects of childhood trauma on food insecurity in overall models,

whereas this effect did not persist when models were split by community.

Additionally, this study indicates that community level factors should be considered when developing nutritional and mental health interventions. Many communities in South Africa are still dealing with the long-term effects of apartheid; this may have a continued effect on stress and mental health in these communities.[53] However, racial disparities exist globally affecting physical and mental health in specific communities differentially to others.[54 55] Specifically, in targeting mental health, contextual factors such as differences in stigma to accessing care, gender norms affecting agency or education levels for women may have significant differential effects within communities. This community context may be important to understand how to best address key risk factors for food insecurity and to inform design of effective interventions.

### Strengths and limitations
The inclusion criteria for the parent study were broad to ensure generalisability. However, recruitment was done during antenatal care visits, so mothers who did not present for antenatal care or who presented in their third trimester were excluded, which may affect overall generalisability. Furthermore, generalisability may be limited to similar population groups, specifically pregnant mothers and similar communities. In addition, approximately 200 mothers who were enrolled in the study were not included in the analysis due to incomplete data. While it is possible that this subset of mothers is at higher risk for many of the factors investigated, there were not significant differences in key factors investigated for mothers included versus those excluded in the current study. As the current study included cross-sectional data, we cannot assert the direction of causality. While a strength of this study was its investigation of the individual impact of different subtypes of IPV, too few mothers experienced or reported sexual IPV to allow this to be investigated fully. Finally, as all variables used were based on self-report, mothers may have under-reported both risk factors and food security due to social desirability bias. Despite these limitations, the current study provides a novel quantitative analysis with a large sample size conducted in an LMIC. Our findings corroborate previous research on risk factors for food insecurity and build on unpublished data in this cohort by investigating subtypes of IPV as well as maternal depression as a mediator for both childhood trauma and IPV.

### CONCLUSIONS
Addressing depression during pregnancy through screening and referral services may help to alleviate the negative impact of IPV, childhood trauma and depression on food security; though direction of causality cannot be asserted by the current study, significant associations between these variables and food insecurity were found. Both IPV victimisation and experiencing

childhood trauma were associated with depressive symptoms in mothers, after controlling for maternal income and education. This may impact their household managerial skills by decreasing motivation to obtain food, to find and hold employment and through decreased physical and cognitive functioning. Our findings highlight the importance of comprehensive programmes aimed at nutrition support or food security, as well as the importance of addressing multiple concurrent psychosocial risk factors that may help to reduce food insecurity and alleviate its negative impact on child health. However, notably, mediation models indicate that the effects of maternal mental health issues are experienced differently at the two communities in this cohort highlighting the importance of programme and policy efforts targeted to specific community profiles. Notably, though not the focus of the current study, both maternal income and maternal education were highly correlated with food insecurity—education especially is likely an important factor cross cutting all key risk and outcome variables—promoting educational opportunities represents an important intervention to improve maternal, and thus, child health.

**Author affiliations**
$^1$Department of Paediatrics and Child Health, Red Cross War Memorial Children's Hospital, University of Cape Town, Cape Town, South Africa
$^2$South African Medical Research Council Unit on Child & Adolescent Health, Cape Town, South Africa
$^3$Department of Behavioral and Social Sciences and International Health Institute, Brown University School of Public Health, Providence, Rhode Island, USA
$^4$Department of Behavioral and Social Sciences and Center for Alcohol and Addiction Studies, Brown University, Providence, USA
$^5$Department of Psychiatry and Mental Health, University of Cape Town, Cape Town, South Africa
$^6$South African Medical Research Council Unit on Risk and Resilience in Mental Disorders, Cape Town, South Africa
$^7$Division of Developmental Paediatrics, Department of Paediatrics & Child Health, Red Cross War Memorial Children's Hospital, University of Cape Town, Cape Town, South Africa

**Acknowledgements** We would like to thank the study staff in Paarl, the study data team and lab teams, the clinical and administrative staff of the Western Cape Government Health Department at Paarl Hospital and at the clinics for support of the study. We acknowledge the advice from members of the study International Advisory Board and thank our collaborators. We would like to thank the families and children who participated in this study.

**Contributors** WB and JP conceptualised the analysis and wrote the first draft of the manuscript. HJZ is principal investigator of the parent study; DS leads the psychosocial study aspects; NK and KD are coinvestigators and contributed to the study design and implementation. JP conducted the data analysis. CK, JP and DS provided critical inputs on the manuscript. All authors read and approved the final manuscript.

**Funding** The study was funded by the Bill and Melinda Gates Foundation (OPP 1017641) with support from the National Institutes of Health (H3Africa 1U01AI110466-01A1). Additional support for DJS, NK, WB and HJZ, and for research reported in this publication was from the South African Medical Research Council (SAMRC) and the UK Royal Society, Newton Advanced Fellowship. WB is supported by the SAMRC National Health Scholars programme; NK receives support from the SAMRC under a Self-Initiated Research Grant.

**Disclaimer** The views and opinions expressed are those of the authors and do not necessarily represent the official views of the SAMRC.

**Competing interests** None declared.

**Patient consent for publication** Not required.

**Ethics approval** Ethical approval was obtained from the Faculty of Health Sciences Research Ethics Committee, University of Cape Town (401/2009) and the Provincial Research Committee.

**Provenance and peer review** Not commissioned; externally peer reviewed.

**Data sharing statement** No additional data are available. All study instruments used in this analysis are available to requesting researchers.

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
