## [Reviewer comments · BMJ Open]

ARTICLE DETAILS

TITLE (PROVISIONAL)	Maternal depression as a mediator for psychosocial risk factors of food insecurity among pregnant women in a South African setting.
AUTHORS	Barnett, Whitney; Pellowski, Jennifer; Kuo, Caroline; Koen, Nastassja; Donald, Kirsten; Zar, Heather; Stein, Dan

VERSION 1 - REVIEW

REVIEWER	Dr Linda Murray University of Tasmania, Australia
REVIEW RETURNED	12-May-2018

GENERAL COMMENTS	Thank you for the opportunity to review this interesting paper. I have the following comments: Background: It might be worth differentiating between the types of middle income countries as there is great variation within these countries. Perhaps Low and lower middle income (LALMI) countries is more descriptive. P3 line 47 - change "little investigated" to "rarely investigated". P3 line 50 - it would be good to briefly outline the effects of food insecurity on infant and maternal wellbeing briefly here (I know they are described earlier, but briefly state again). Methods: p4 line 11/12 - can you say something about the levels of education and employment within this community, and the mix of ethnicities? p4: 37-38 - was there a cut off used for the EPDS? You seem to imply it was measured as a continuous variable, but at what score would you consider depression to be occurring (even at a mild level?). This is particularly important as later on you present depression as a dichotomous variable - indicating that you used a cutoff? Please describe this measure more clearly. Description of cutoffs also needs to occur for other psychometric measures used (i.e the information under Table 1 needs to also be presented here in the methods section). On page 6, your table needs to be labelled more clearly. Are all figures given just the total N? Please make this clear, and also consider presenting percentages. - The discussion reads well - perhaps you could expand a little more on the implications of the results in regards to designing programs for different communities. A pleasure to read.
---

REVIEWER	Ellen Poleshuck University of Rochester Medical Center USA
REVIEW RETURNED	25-Jul-2018

GENERAL COMMENTS	This interesting study tackles an important topic and investigates what is associated with food insecurity among pregnant women. The large sample size, rigorous measures, important content, and strong writing are all strengths of the paper. Specific feedback for improvement is as follows:  -There does not seem to be a clear theoretical model or rationale provided for the variables selected for this study, and why depression would be expected to mediate the relationship between intimate partner violence (IPV) and childhood trauma and food scarcity. A stronger conceptual rationale needs to be provided. -Relatedly, it is interesting that the authors define intimate partner violence, childhood trauma, and stressful life events as mental health problems. While these experiences are highly associated with both physical and mental health, it seems problematic to assume that experiencing trauma or abuse is the equivalent of a mental health problem. -How was the definition of food insecure determined (two or more items endorsed)? -Why was childhood trauma and stressful life events entered in Block 2 instead of Block 1 with IPV or as a separate block? -The Hernandez et al and Sun et al articles should be referenced in the introduction as well as the discussion since in some ways this study attempts to replicate their work. -A study limitation is that the generalizability of the findings to other samples is unknown. -The authors need to be more cautious in their language and conclusions given that this is a cross-sectional study. They cannot suggest that they were able to “predict” food scarcity. The conclusions paragraph seems to go beyond the results provided by this study.
---

REVIEWER	Dr Brian Kelly Bradford Institute for Health Research Bradford, BD9 6RJ England, UK
REVIEW RETURNED	30-Jul-2018

GENERAL COMMENTS	I have been asked to review and comment on the statistics section of the study. The statistical analysis employed is well set out in the 'methods' section of the paper, it is welcome that this covers both the statistical methods used, and how the modelling strategy employed relates to answering the research questions. The results of the statistical analysis are generally well presented.
--

REVIEWER	Sera Young Northwestern University, USA
REVIEW RETURNED	08-Aug-2018

GENERAL COMMENTS

Thanks for the opportunity to review this interesting paper which seeks to understand how maternal mental health and food insecurity covary.

It took several readings for me to understand the messages being conveyed. Here are some ideas for clarity:

I found the title to be confusing and vague. This is perhaps tedious, but much more clear: The association between IPV & FI and childhood trauma and FI are mediated by maternal depression.

Including a conceptual framework, without data, of how the various psychosocial risk factors relate to FI to orient the reader in the Introduction.

A table of assessments would help reader, with columns to include name of measurement, how modified, potential range, cut-offs/operationalization if applicable, and reference/s. It's very important to be clear on adaptation, e.g. how was the USDA FI score modified?

I'm confused about study design. Were there only two visits in the entire study? If there are two, is that still cross-sectional?

The authors could better work on concisely situating their work in the context of that which has been done already, i.e. concurrent measurement of a number of psychosocial constructs in LMIC, and/but how much further does this paper take the analyses that have already been done with data from this study? Explanations of what this adds, in intro and discussion, are a bit clunky.

Data on statistical differences between women who missed 1 of 2 visits should be made available at least in OSM.

Analytically, I'm not sure why data are presented by community, rather than outcomes or predictors of interest. For those not in this area, it is not nearly as interesting to know how statistical relationships worked by site as it is to know how associations differed by, e.g. level of depression, IPV, or FI. Table 1 with bivariate data on FI by all these characteristics seems like it would advance the science more. You could control for community as a fixed effect in subsequent analyses.

Why not group mediation analyses by community into a single, multi-panel figure (Fig 1a-1c), so that the reader can compare values for each pathway.

Please perform power calculations.

Did you explore sexual IPV? Why do you think sexual IPV was not associated with FI? (Probably because it was less frequently reported and therefore underpowered?)

In the conclusion, there could be more of an acknowledgement to the factors which are strong predictors of food insecurity which were controlled for. Please discuss how that could affect your observations.

	Smaller: Include scale for measurement of FI in abstract. Second sentence—unable to meet dietary requirements is not FI. Asthma is also a medical condition. What is meant by a stable community? In the background section, the third paragraph (line 30, page 2) consider splitting into two paragraphs: one focused on IPV and one focused on maternal childhood trauma. Please consider making study instruments available as online supplementary material. Was this study registered in a clinical trials repository? The sentence about “severely hungry school-aged children” (line 36-37, page 3) should be omitted or reworded, it seems out of place where it is currently placed in this paragraph and it muddles the fact that this article is focused on maternal factors/experiences and not those of the children. What are examples of stressful or negative life events as characterized and evaluated by SASH standards? (Line 46, page 4) Explain social grants. (line 46, page 5) Table 1—how is depression, IPV, etc operationalized? Page 6, subheading. These are bivariate analyses. There were some redundancies in the Introduction and Discussion. I’m not seeing rationale for sample size. Sample sizes should be included in all figures. Throughout, the authors need to be careful with causal language, since this is a cross-sectional analysis. In the discussion section, explain how lines 3-6 on page 9 may be linked to traditional gender roles. Line 28-30, page 9 – Is there any evidence that intergenerational trauma counselling alleviates effects of food insecurity? Line 50, page 9 – social support was never mentioned before this point, why now and not previously – is there evidence of better social support in Mbekweni? Was social support measured? Super tiny: Line 5, page 2 – comma after “low infant birthweight” Line 10, page 4 – “comprising” □ “comprised of”; remove comma after “200 000 people”
--	--

VERSION 1 – AUTHOR RESPONSE

Reviewer 1 comments:

6. Background: It might be worth differentiating between the types of middle income countries as there is great variation within these countries. Perhaps Low and lower middle income (LALMI) countries is more descriptive. Thank you for this suggestion, we have now included as low and middle income countries, as South Africa is a middle income country.

7. P3 line 47 - change "little investigated" to "rarely investigated". This has been changed.

8. P3 line 50 - it would be good to briefly outline the effects of food insecurity on infant and maternal wellbeing briefly here (I know they are described earlier, but briefly state again). Thank you we have now added the following to replace wellbeing with specific outcomes: "Given the long-term health implications of food insecurity for child development as well as maternal and child mental and physical health."

Methods:

9. p4 line 11/12 - can you say something about the levels of education and employment within this community, and the mix of ethnicities? Thank you for suggestion, we have added the following to the text to provide prevalence of some key community indicators: "It is a low socioeconomic community comprising approximately 200 000 people predominantly of mixed-ethnicity (62.5%; 13.5% caucasian; 22.7% Black African). The district is characterized by a high prevalence of a range of health risk factors such as single-parent households, depression, childhood trauma, IPV, poverty, low levels of education (27.4% completing secondary school) and high unemployment (17.6%)."

10. p4: 37-38 - was there a cut off used for the EPDS? You seem to imply it was measured as a continuous variable, but at what score would you consider depression to be occurring (even at a mild level?). This is particularly important as later on you present depression as a dichotomous variable - indicating that you used a cutoff? Please describe this measure more clearly. Description of cutoffs also needs to occur for other psychometric measures used (i.e the information under Table 1 needs to also be presented here in the methods section). Thank you for this suggestion – we have included cut off scores in the methods section for all relevant variables, including citations for determination of cut off value. Cut off scores for depression are used to describe the data (Table 1) whereas continuous scores are used for hierarchical and mediation analyses. This is now noted at the bottom of Table 2.

11. On page 6, your table needs to be labelled more clearly. Are all figures given just the total N? Please make this clear, and also consider presenting percentages. Total n as well as percentages are presented. We have labeled to ensure clarity.

12. The discussion reads well - perhaps you could expand a little more on the implications of the results in regards to designing programs for different communities. A pleasure to read. Thank you, we have expanded upon this in the discussion section, specifically adding the following paragraph: "Additionally, this study indicates that community level factors should be considered when developing nutritional and mental health interventions. Many communities in South Africa are still dealing with the long-term effects of apartheid, this may have a continued effect on stress and mental health in these communities. However, racial disparities exist globally affecting physical and mental health in specific communities differentially to others. Specifically, in targeting mental health, contextual factors such as differences in stigma to accessing care, gender norms affecting agency or education levels for women, may have significant differential effects within communities. This community context may be important to understand how to best address key risk factors for food insecurity and to inform design of effective interventions."

Reviewer 2 comments:

13. There does not seem to be a clear theoretical model or rationale provided for the variables selected for this study, and why depression would be expected to mediate the relationship between intimate partner violence (IPV) and childhood trauma and food scarcity. A stronger conceptual rationale needs to be provided. Thank you – we have added a figure to assist the reader with introducing the conceptual framework. Further, we have re-organised the introduction so that the rationale for the manuscript is better articulated.

14. Relatedly, it is interesting that the authors define intimate partner violence, childhood trauma, and stressful life events as mental health problems. While these experiences are highly associated with both physical and mental health, it seems problematic to assume that experiencing trauma or abuse is the equivalent of a mental health problem. Thank you – an important point. We have edited the text to ensure trauma/violence exposures are appropriately described as such rather than implying that they represent mental health problems.

15. How was the definition of food insecure determined (two or more items endorsed)? Correct, if two or more items were endorsed by mothers antenatally, households were categorised as food insecure, this was based on guidelines from the USDA regarding its short form Food Security Module. Additional details regarding the FI assessment has been added to the manuscript.

16. Why was childhood trauma and stressful life events entered in Block 2 instead of Block 1 with IPV or as a separate block? Thank you for this comment, we definitely agree that given the research question, including IPV, childhood trauma and stressful life events should be included in the same block with mental health variables included separately. Table 2 has been altered to present these results more clearly, linked it to the hypothesis and the conceptual framework is also now included.

17. The Hernandez et al and Sun et al articles should be referenced in the introduction as well as the discussion since in some ways this study attempts to replicate their work. Thank you, this has been added to the introduction and adds valuable background context to our study.

18. A study limitation is that the generalizability of the findings to other samples is unknown. Thank you, we have amended the discussion to acknowledge the fact that our findings may not be generalisable to similar populations and communities.

19. The authors need to be more cautious in their language and conclusions given that this is a cross-sectional study. They cannot suggest that they were able to “predict” food scarcity. The conclusions paragraph seems to go beyond the results provided by this study. We have included a statement in conclusions indicating that causality is unknown based on the current study and highlighting that only associations between key variables were found.

Reviewer 3 – no changes requested

Reviewer 4 comments:

20. It took several readings for me to understand the messages being conveyed. Here are some ideas for clarity: I found the title to be confusing and vague. This is perhaps tedious, but much more clear: The association between IPV & FI and childhood trauma and FI are mediated by maternal depression. Thank you, we have amended the title for clarity to “Food insecure pregnant women in South Africa: Maternal depression mediates violence and trauma risk factors.”

21. Including a conceptual framework, without data, of how the various psychosocial risk factors relate to FI to orient the reader in the Introduction. Thank you for this suggestion – we have included a conceptual framework as Figure 1.

22. A table of assessments would help reader, with columns to include name of measurement, how modified, potential range, cut-offs/operationalization if applicable, and reference/s. It's very important to be clear on adaptation, e.g. how was the USDA FI score modified? Thank you, we have decided to rather include cut off scores and potential ranges in the methods section as there are quite a few tables/figures already. This and clarity regarding the FI score has been added in text.

23. I'm confused about study design. Were there only two visits in the entire study? If there are two, is that still cross-sectional? The parent cohort study is ongoing with two antenatal visits and multiple postnatal visits through at least child age 5 years. The current paper focuses on data collected at the two antenatal visits; though the parent study is longitudinal, the key variables included in this analysis were each measured only once antenatally, we therefore consider this a cross-sectional analysis. We have modified the text for clarity.

24. The authors could better work on concisely situating their work in the context of that which has been done already, i.e. concurrent measurement of a number of psychosocial constructs in LMIC, and/but how much further does this paper take the analyses that have already been done with data from this study? Explanations of what this adds, in intro and discussion, are a bit clunky. We have amended text in the introduction and discussion to better expand upon the knowledge gaps the current study aims to address.

25. Data on statistical differences between women who missed 1 of 2 visits should be made available at least in OSM. All mothers with antenatal psychosocial data and food security data were included in the present analysis; we are therefore only able to include a comparison of sociodemographic variables. This has been added as a supplementary table as well as in text.

26. Analytically, I'm not sure why data are presented by community, rather than outcomes or predictors of interest. For those not in this area, it is not nearly as interesting to know how statistical relationships worked by site as it is to know how associations differed by, e.g. level of depression, IPV, or FI. Table 1 with bivariate data on FI by all these characteristics seems like it would advance the science more. You could control for community as a fixed effect in subsequent analyses. Thank you, we have chosen to present analyses split by community because of the large differences, socioeconomic, cultural, clinical and psychosocial between the two communities that could have significant bearings on the results of the mediation models. Given this focus for the mediation models, we have therefore kept table 1 comparing key variables by study community. We have added the rationale for this approach into the statistical analysis section of the manuscript.

27. Why not group mediation analyses by community into a single, multi-panel figure (Fig 1a-1c), so that the reader can compare values for each pathway. Thank you for this suggestion, we agree this will make it easier to compare mediation models; we have done so for all.

28. Please perform power calculations. The current study is nested in a larger birth cohort study. Power calculations were done for the parent study, which informed recruitment of the study sample; therefore sample size was calculated prior to conception of the current article. As a secondary data analysis all participants with available data were included in the current article.

29. Did you explore sexual IPV? Why do you think sexual IPV was not associated with FI? (Probably because it was less frequently reported and therefore underpowered?) This is correct, too few women reported sexual IPV exposure (n=68/992), therefore we believe that the analysis was underpowered to detect an association. This is included as a limitation in the discussion section.

30. In the conclusion, there could be more of an acknowledgement to the factors which are strong predictors of food insecurity which were controlled for. Please discuss how that could affect your observations. Thank you, we have included an acknowledgement of the key sociodemographic variables that were highly correlated with food insecurity and have added this as an important consideration in the conclusion.
31. Include scale for measurement of FI in abstract. Thank you, this has been added to abstract.
32. Second sentence—unable to meet dietary requirements is not FI. Thank you, have corrected and updated cited figures.
33. Asthma is also a medical condition. This has been corrected.
34. What is meant by a stable community? This is to indicate that the community has relatively little in and out migration (compared to other communities in South Africa); we have actually decided to remove this as confusing and not critical to state for the current analysis.
35. In the background section, the third paragraph (line 30, page 2) consider splitting into two paragraphs: one focused on IPV and one focused on maternal childhood trauma. Thank you, we have edited this text, though have decided to leave these in a single paragraph.
36. Please consider making study instruments available as online supplementary material. Thank you, we have added a statement that all study instruments are available upon request; almost all are based on commonly used, validated measures that are widely available.
37. Was this study registered in a clinical trials repository? The parent study is registered with H3 Africa (NIH) Biorepository, some data has been deposited, however data collection is ongoing. The core study is dedicated to data sharing and is involved in many collaborations with external investigators; our data sharing provisions are based on proposals from external investigators or groups and at the discretion of the PI.
38. The sentence about “severely hungry school-aged children” (line 36-37, page 3) should be omitted or reworded, it seems out of place where it is currently placed in this paragraph and it muddles the fact that this article is focused on maternal factors/experiences and not those of the children. Thank you, this has been removed and background section re-phrased.
39. What are examples of stressful or negative life events as characterized and evaluated by SASH standards? (Line 46, page 4) Examples of these have been added to the methods section (specific examples include: serious illness, major financial crisis, serious discord with family or friends).
40. Explain social grants. (line 46, page 5) An explanation for social grants (receiving government support for child care or disability) has been added to the methods section.
41. Table 1—how is depression, IPV, etc operationalized? Thank you, details of psychosocial risk factors where dichotomised, have been included in the methods section of the text as well as in footnote for Table 1.
42. Page 6, subheading. These are bivariate analyses. Thanks, this has been corrected throughout.
43. There were some redundancies in the Introduction and Discussion. Thank you, we have reviewed the text and made edits with this in mind.
44. I’m not seeing rationale for sample size. Please see response to item 23 – essentially the current analysis is a nested sub-study of a birth cohort study; sample size for the parent study was calculated prior to conception of the current article.

45. Sample sizes should be included in all figures. These have been added.
46. Throughout, the authors need to be careful with causal language, since this is a cross-sectional analysis. Thank you, we have edited the text with this comment in mind, with a particular focus on the conclusions section.
47. In the discussion section, explain how lines 3-6 on page 9 may be linked to traditional gender roles. Thank you, we have expanded upon this point.
48. Line 28-30, page 9 – Is there any evidence that intergenerational trauma counselling alleviates effects of food insecurity? Though counseling is often cited as a potential intervention e.g. in conclusion of articles, it is most often cited for alleviating associations between trauma/violence exposure and negative mental health outcomes, we cannot find literature investigating counselling interventions on food security. We have indicated that this may be a direction for future research.
49. Line 50, page 9 – social support was never mentioned before this point, why now and not previously – is there evidence of better social support in Mbekweni? Was social support measured? Social support is only measured in the cohort at 3 years postnatally, so not known for the current analysis/time points. We have, however, added references to previous studies, one in a similar community in SA where social support was found to buffer depressive symptoms.
50. Line 5, page 2 – comma after “low infant birthweight” This has been corrected.
51. Line 10, page 4 – “comprising” \ “comprised of”; remove comma after “200 000 people” This has been corrected.

VERSION 2 – REVIEW

REVIEWER	Brian Kelly Bradford Institute for Health Research United Kingdom
REVIEW RETURNED	18-Dec-2018

GENERAL COMMENTS	I think the authors have generally responded well to the reviewer comments. However, I am not sure if the amended title quite achieves what was requested by the editors, as it still makes no reference to study design. Apart from that I would recommend acceptance.
--